

# Zbrowse: an interactive GWAS results browser

Greg R. Ziegler[1], Ryan H. Hartsock[2] and Ivan Baxter[1,2]

[1] United States Department of Agriculture — Agricultural Research Service, St. Louis, MO, USA
[2] Donald Danforth Plant Science Center, St. Louis, MO, USA

## ABSTRACT

The growing number of genotyped populations, the advent of high-throughput phenotyping techniques and the development of GWAS analysis software has rapidly accelerated the number of GWAS experimental results. Candidate gene discovery from these results files is often tedious, involving many manual steps searching for genes in windows around a significant SNP. This problem rapidly becomes more complex when an analyst wishes to compare multiple GWAS studies for pleiotropic or environment specific effects. To this end, we have developed a fast and intuitive interactive browser for the viewing of GWAS results with a focus on an ability to compare results across multiple traits or experiments. The software can easily be run on a desktop computer with software that bioinformaticians are likely already familiar with. Additionally, the software can be hosted or embedded on a server for easy access by anyone with a modern web browser.

## INTRODUCTION

The recent development of high-throughput phenotyping techniques coupled with the ability to genotype large populations of diverse individuals has revolutionized the way that forward genetics research is performed. Tools have rapidly become available to perform genome-wide association studies (GWAS) in a variety of species (*Kang et al., 2010*; *Segura et al., 2012*; *Lipka et al., 2012*) that can map traits to the genome with high enough resolution to quickly provide a tractable list of potential causal genes.

One of the first steps an analyst will take is determining what gene or genes fall under the SNP peaks that can be seen on the Manhattan plot. Unfortunately, these plots are generally not interactive. Identifying the peaks of interest usually involves sifting through the results table for the range of coordinates under the peak of interest and then using those coordinates to filter a large gene annotation file. The extra steps involved in exploring the data in this way makes it more likely that interesting associations may be missed either due to (1) mistakes made in attempting to mine the large results files or (2) the dataset not being mined deeply enough due to the difficulty of looking for genes under less significant peaks. Additionally, this method quickly becomes tedious when analyzing multiple phenotypes or relatively complex traits.

Some web applications provide tools for viewing Manhattan plots (Table 1), but they are all either specific to a single species or don't allow interactive results browsing. These

Corresponding author
Ivan Baxter,
ivan.baxter@ars.usda.gov

**Table 1** **Comparison of ZBrowse with other available GWAS visualization software packages.** LocusZoom (*Pruim et al., 2010*), LocusTrack (*Cuellar-Partida, Renteria & MacGregor, 2015*), GWAS Central (*Beck et al., 2014*), GWAPP (*Seren et al., 2012*), GWASrap (*Li, Sham & Wang, 2012*), JBrowse (*Skinner et al., 2009*).

| Browser | Language programmed in | Is it run on local, server or web | Is the plot interactive | Display SNPs and intervals | Organisms supported | Displays multiple GWAS experiments |
|---|---|---|---|---|---|---|
| LocusZoom | R, Python wrapper, SQLite table | Local or web | No | No | Human | No |
| LocusTrack | R | Local or web | No | No | Human | No |
| GWAS Central | Unclear | Web | Partial | No | Human | No |
| GWAPP | Python, HTML5, Javascript | Web | Yes | No | Arabidopsis | No |
| GWASrap | Unclear | Web | Yes | No | Human | No |
| JBrowse | Perl/Javascript | Local, server or web | Yes | Yes | Any | Not easily on the same track |
| ZBrowse | R | Local, server with some modification | Yes | Yes | Any | Yes |

resources also do not allow for easy viewing and comparison of GWAS results across phenotypes and studies, a situation that frequently arises with structured populations.

## GOALS

We approached the construction of a new GWAS browser with the goal of giving the users the following tools, all of which were focused on versatility and adaptability:

1. *Ability to plot multiple traits in the same panel.* We wanted to enable users to find genotype-environment (GxE) interactions (e.g., those instances where an environmental condition causes a phenotypic effect, but only for individuals with a given allele) and loci with pleiotropic effects (the same loci affecting multiple phenotypes).

2. *Ability to rapidly move between scales (thousands of bps to billions).*

3. *Ability to find overlaps or commonalities among datasets.*

4. *Ability to interact directly with the plots.* We wanted the ability to look at the annotations of genes inline easily and link to additional information.

5. *Ability to plot both SNPs and genetic intervals.* We wanted users to be able to combine the results of quantitative trait locus mapping techniques with GWAS results.

6. *Ability to download plots and gene lists.*

7. Finally, we wanted all of this information and functionality to be available in one browser window using tools that are common and freely available on most personal computers.

Here, we present an interactive GWAS results viewer that is an extension of the classic GWAS Manhattan plot. It allows for the rapid comparison of GWAS results from multiple phenotyping experiments and the rapid viewing and analysis of genes under peak SNPs. *Arabidopsis thaliana*, maize, soybean, and sorghum are bundled with the software but we provide instructions and tools to easily add support for other organisms, including

those with sex chromosomes. As a practical application of the browser's usage, we will demonstrate it using results from a recently completed sorghum GWAS experiment in which elemental profiling phenotypes were measured in accessions grown across three separate locations (*Shakoor et al., 2015*).

## USER INTERFACE

The ZBrowse GWAS results viewer is an interactive application that runs on a local machine using R and is rendered in any modern web browser. Because the browser runs on the user's local machine, the data can remain private. Though the focus of the first version is a local installation, the browser display allows for easy sharing of the application. The browser is designed to be a streamlined environment that provides fast access to visualization tools for GWAS results. ZBrowse utilizes a tab-based navigation format to make accessing different aspects of the browser fast, efficient, and intuitive. There is also a sidebar panel on the left of the page that updates with a set of options specific to the tab being displayed.

The first tab in the list, and the landing page when the application is first loaded, is the Manage tab (Fig. 1). This tab allows a new GWAS dataset to be uploaded into the application or a pre-loaded dataset from a dropdown menu can be selected. Data can be uploaded in a flat file delimited with either commas or tabs or an RData object. These flexible file formats allow any type of data to be loaded into the browser.

In Fig. 1, we have loaded the results from the sorghum ionomics experiment and selected the appropriate columns to be used for plotting the results. The results file was generated by taking the most significant SNP hits from each of the 80 GWAS experiments performed (20 phenotypes measured in 3 locations and an experiment combining the location data). We added a column describing which experiment (e.g., the three locations) and which phenotype each SNP was found in.

Once uploaded, a preview of the first ten rows of the dataset will appear in the main panel. Below this table is a series of selection boxes that allow the user to specify which columns in the file to use for plotting. This selection method removes the complexity of requiring the input file to either have columns with specific names or columns in a specific order.

The user needs to select a chromosome and base pair for determining the location of each SNP in the genome. To plot base pair intervals, there is a separate checkmark box. Checking this box opens a second set of selection boxes allowing the user to select columns defining a start base pair and a stop base pair for each interval, as well as a separate $y$-axis column (allowing the traits to be plotted on different scales). Chromosomes expected for each organism are defined in a separate file (See Adding Organisms section). The browser supports both numeric and alphabetical chromosome names, so scaffold names (i.e., scaffold_1, scaffold_2), sex chromosomes (i.e., X, Y), or chromosome arm designations (i.e., 1S, 1L) are acceptable.

If the uploaded dataset is data from only one GWAS trait, there is a checkbox to include all data as one trait. Otherwise, the user can select one or more columns to group the

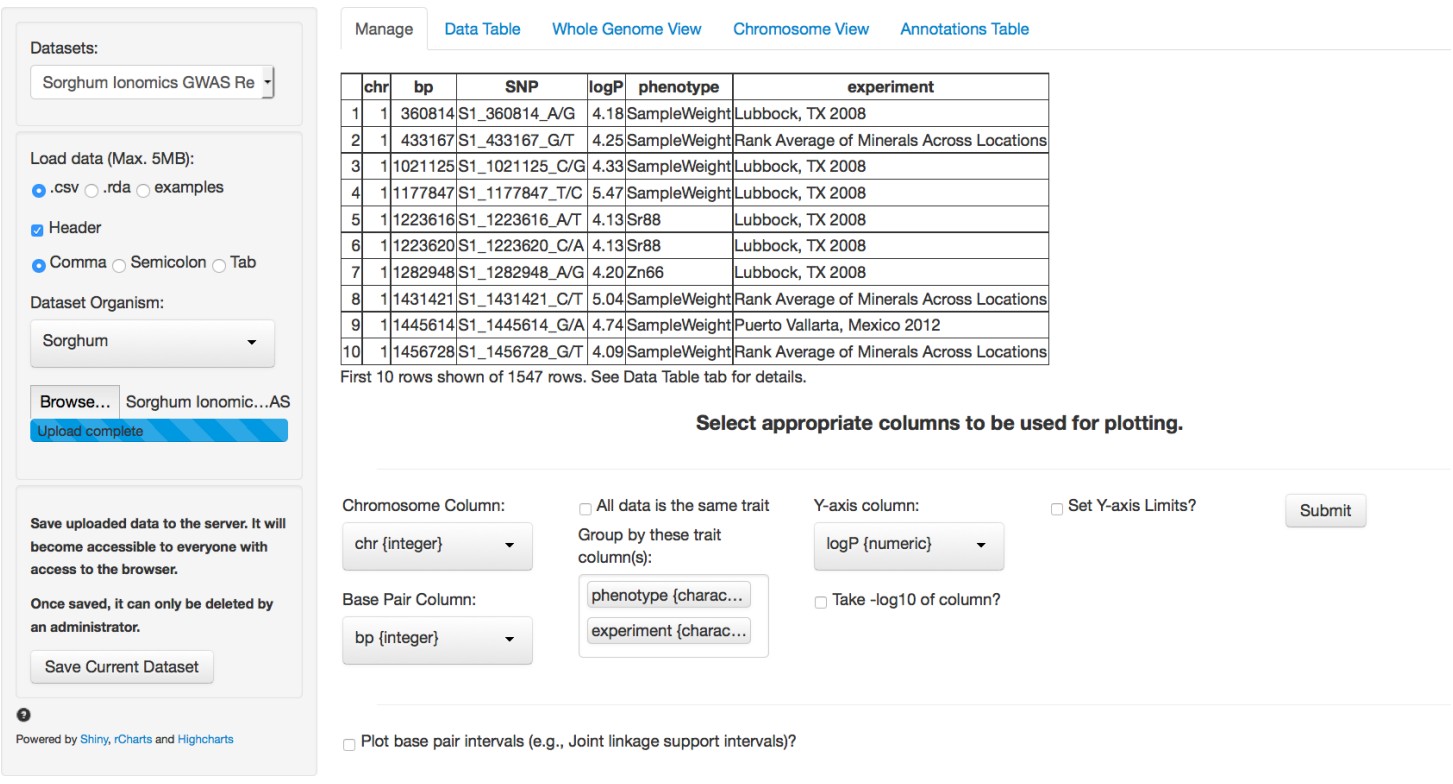

**Figure 1** **Landing page for ZBrowse.** A sorghum GWAS dataset has been uploaded and the appropriate columns for plotting the data have been selected from the selection boxes.

data by when plotting. For example, a researcher might be interested in comparing GWAS results from multiple experiments, or in comparing results from multiple traits measured in the same experiment, or both. The user can designate as many columns as desired to subset the data that will be used to create tracks designated by different colors on the plots. Color coding in the plots for each track will match between the intervals and the single points. In our sorghum dataset, we are interested in exploring relationships between both the phenotypes measured and the effect environment may have had on our GWAS results. Therefore, we have selected the two columns that we added describing what location and phenotype from which each SNP result is derived.

Finally, the user needs to select the *y*-axis column with the significance value against which to plot each SNP. Usually, this is the negative logarithm of the *P*-value, but can also be the number of bootstrap models that include this SNP (RMIP, *Valdar et al., 2009*) or any other measure of trait significance, such as effect size. If a user hasn't already created a column with the negative logarithm of the *P*-value, the *P*-value column can be selected and there is a checkbox to have the software automatically perform the calculation. The final parameter allows for user selectable values for the *Y*-axis scale. By default, the software will automatically scale the *y*-axis based on the range of the selected data. The browser will only display 5,000 points total (see "Limitations" section). If there are more than 5,000 points in

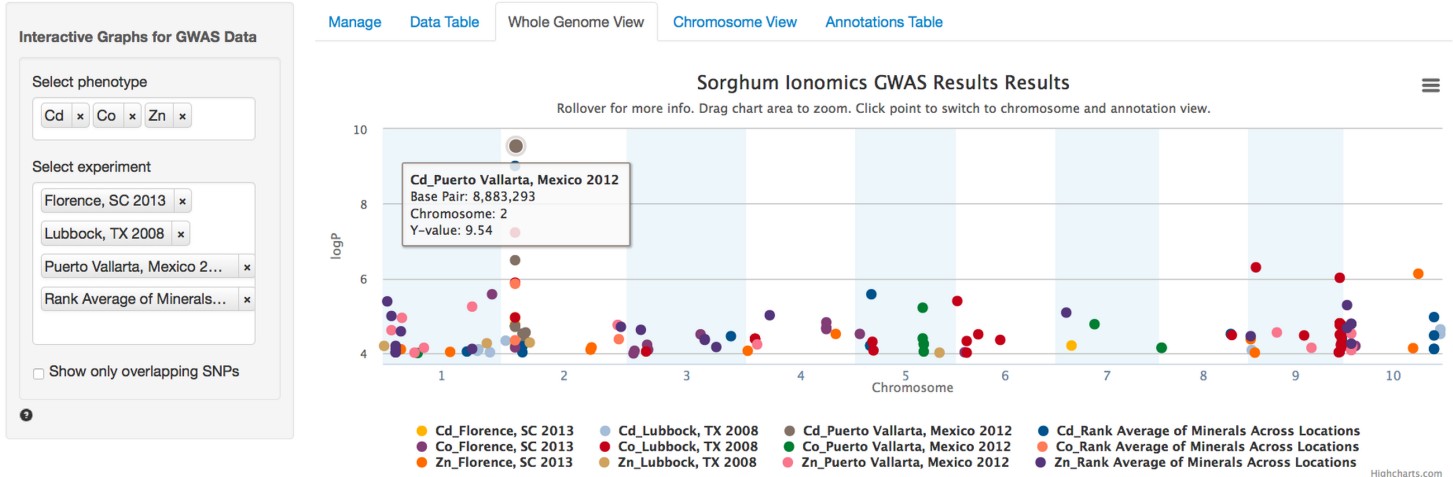

**Figure 2** **Genome wide view of ZBrowse manhattan plot.** The plot is displaying a comparison of GWAS results from three phenotypes measured in three separate locations and one aggregate experiment (12 GWAS experiments total on one graph). The legend at the bottom of the figure displays the color of points that correspond to the combination of traits and locations selected in the sidebar on the left hand side of the figure. Clicking the points in the legend allows a user to easily show or hide points from that trait. The title of the plot is automatically generated from the filename of the dataset provided by the user. This makes it easy to determine which GWAS experiment is being plotted. The tooltip popup is displayed by hovering the mouse over points in the plot.

the subset of tracks being plotted, the browser will use the $y$-axis column to rank the SNPs and take only the top 5,000.

After the user has selected the appropriate parameters, clicking the submit button will trigger a tab change to the Whole Genome View visualization tab (Fig. 2). Conveniently, once submitted, the software will remember the selected settings for this dataset on future visits and automatically populate the fields with the previously selected parameters. The plot on this tab is formatted as a standard genome-wide Manhattan plot. The $x$-axis is ordered by chromosome and base pair within each chromosome. The background of the plot has alternating blue/white shading for the even and odd chromosomes to highlight chromosome breaks. The panel on the left contains a box for each trait column selected in the Manage tab. In the case of our Sorghum ionomics dataset, there is a box where we can select which combination of the 20 phenotypes we would like to plot and which experimental location we would like to plot. In Fig. 2, we have selected all four locations and we are plotting the significant associations found for cadmium, cobalt, and zinc. There is also an option for showing only overlapping SNPs with the ability to adjust both the overlap size around each point and the minimum number of overlaps.

When the user scrolls over points in the plot, a tooltip will display that shows information about the trait that SNP is associated with, the $Y$-axis value, and the exact chromosome and base pair for the SNP (Fig. 2). If the tooltip gets in the way of the viewing or selecting of points, clicking the plot will temporarily hide the tooltip box. Clicking any point in the Whole Genome View will change tabs to the Chromosome View tab with a focus on the clicked point (Fig. 3). In our example, we clicked on the peak SNP for cadmium in the Mexico experiment. This tab contains two plots: one

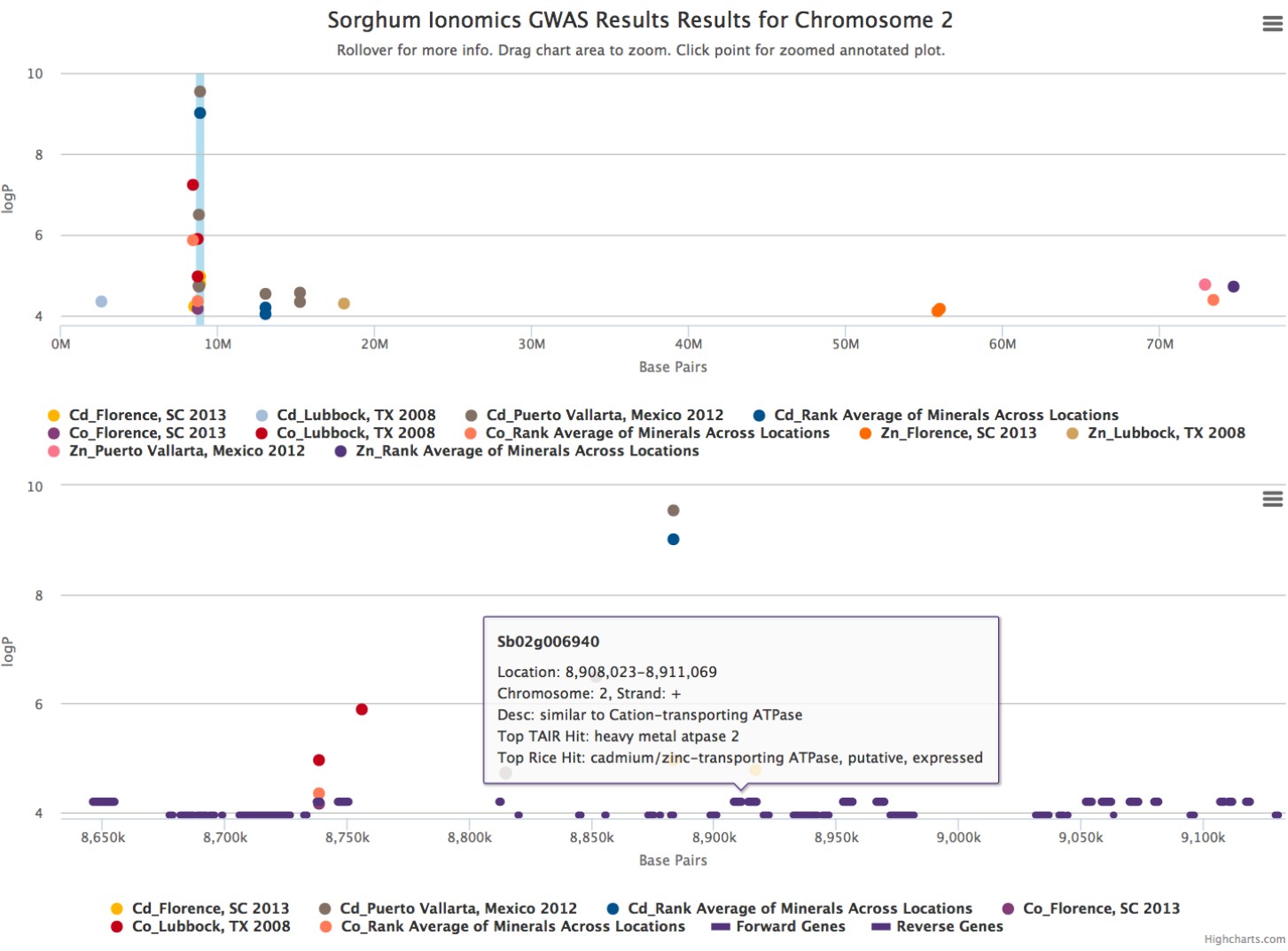

**Figure 3 Chromosome and annotation level manhattan plots in ZBrowse.** This plot was reached by clicking the peak SNP for cadmium in the Puerto Vallarta, Mexico experiment displayed in Fig. 2. The blue vertical bar in the upper chromosome level plot indicates the zoomed region in the annotation level plot below. Two tracks at the bottom of the annotation level plot indicate locations of genes on the forward and reverse strands. Scrolling over these tracks displays a tooltip with a description of the gene and clicking genes in the track opens a separate browser window displaying information about the gene from an external database. The displayed gene, a heavy metal transporter, is a likely candidate for effecting cadmium accumulation in sorghum germplasm.

is a chromosome-wide view displaying the data from the chromosome clicked in the genome-wide view, the other plot is an annotation plot of the region around the clicked base pair. A blue band in the chromosome-wide plot highlights the region being displayed in the annotation plot. In one click, we have gone from viewing the entire 700 million base pair genome, to a plot displaying the region 250 thousand base pairs around our point of interest. The plot contains a variety of interactive features. In addition to selecting traits to view in the sidebar panel, traits can be quickly hidden by clicking their entry in the legend. When many points are plotted on the same graph, overplotting can make it difficult to discern points clustered around the same peak. To alleviate this, the plot can be easily

zoomed by clicking and dragging a zoom box anywhere in the plot. This makes it much easier to see the relationship between tightly grouped points. The displayed chromosome can be changed without returning to the Whole Genome View tab using the drop-down menu in the sidebar panel. Points can be clicked to redraw the annotation plot around new points of interest.

The annotation plot is a variable width plot that defaults to showing the region 250,000 base pairs on either side of the point of interest. The width of this region can be adjusted between 1,000 and 500,000 base pairs using the slider on the sidebar panel. The bottom of this plot has a track that shows the position of coding sequences around the SNP of interest. The tooltip for genes in this track displays information about the gene location, strand, and function, if known. For maize, arabidopsis and soybean, clicking on the gene will open a new browser tab that links to the gene description page specific to the organism being viewed. Arabidopsis links to The Arabidopsis Information Resource (TAIR) (*Lamesch et al., 2011*), soybean links to Soybase (*Grant et al., 2010*), and maize links to the Maize Genetics and Genomics Database (MaizeGDB) (*Lawrence et al., 2004*). In addition, clicking genes in organisms added from Phytozome (*Goodstein et al., 2012*) via the add organism application described below opens the Phytozome description page for that gene. ZBrowse can be easily modified to link out to other species-specific databases that can accept a query string in the URL.

Our cadmium example in Fig. 3 shows how quickly we can find potential leads for candidate genes. Browsing the gene track, a gene almost directly under the peak cadmium SNP is annotated as being similar to a cadmium/zinc-transporting ATPase; clicking this gene opens a new browser tab displaying the phytozome gene description page.

In addition to the visual browser, annotation data can be explored in tabular form in the Annotation Table tab (Fig. 4). This table provides an interactive table of the genes found in the window around the selected point. The table is sortable and searchable and can also be exported as a comma-separated file. A similar table viewer is available in the Data Table tab for analysis of the selected GWAS dataset.

## ADDING ORGANISMS

Currently, maize, soybean, arabidopsis and sorghum are downloaded with the browser source package. We have developed an application to quickly add organisms to the browser from annotations downloaded from the Plant Genomics Portal (Phytozome) to the local installation of ZBrowse. Additionally, we will be formatting requested and popular organisms and releasing the files on GitHub. These will be easy to download and incorporate into your existing browser installation.

Adding a new organism manually requires two additional files to be created and placed into the ZBrowse installation directory. One is a flat text file with three lines. The first line tells the browser the display name for the organism. The second line tells the browser the names and size of each genome feature (i.e., chromosomes, scaffold, etc.) and the third line is the path to a csv file containing the annotation information. The annotation file needs to

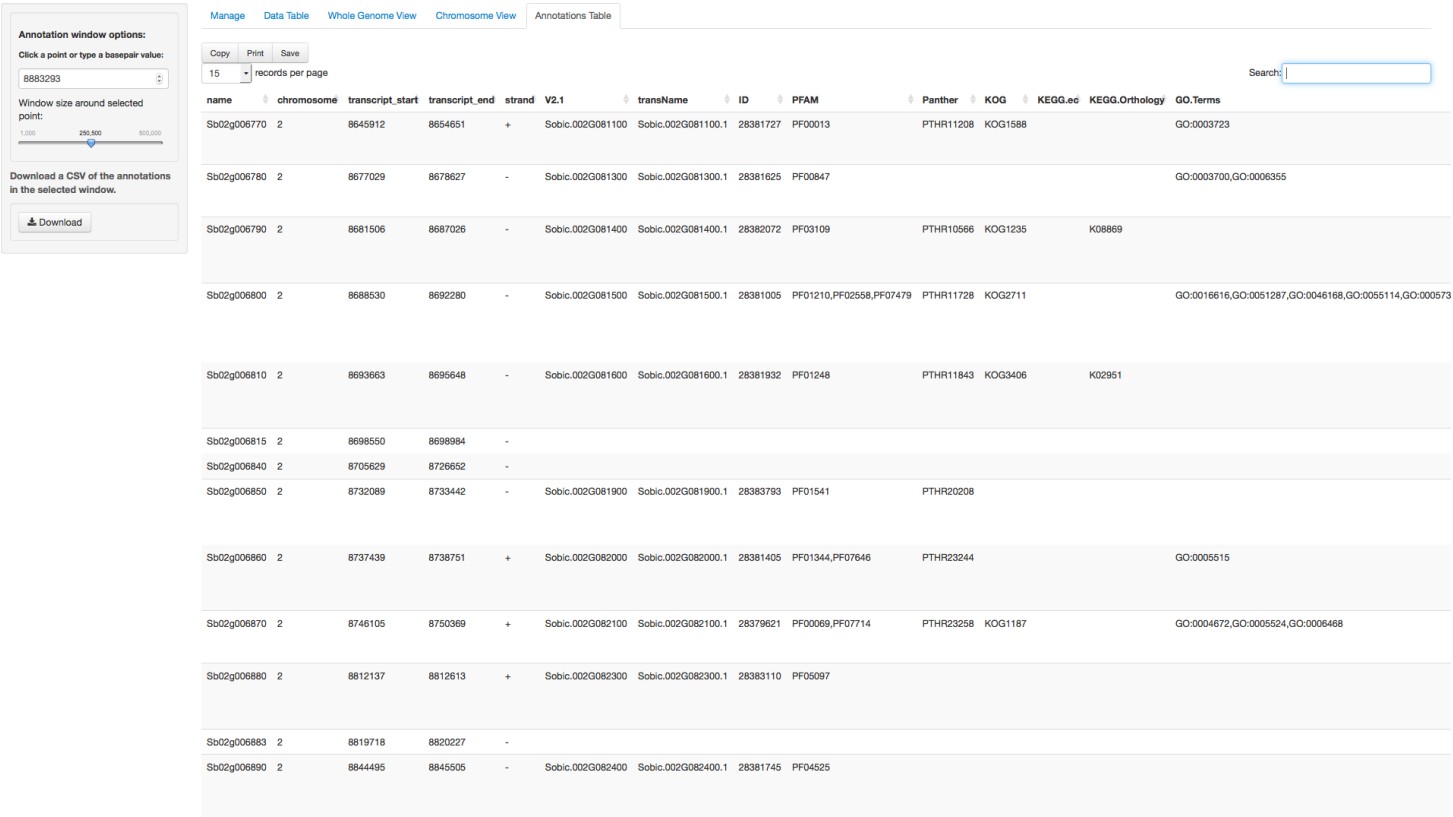

**Figure 4 Annotation Table tab in ZBrowse.** Tabular form of the genes found around a SNP. This table is searchable in the browser and can be exported as a csv formatted file.

have the following columns: name, chromosome, transcript_start, transcript_end, strand, ID, defLine, bestArabhitDefline and bestRiceHitDefline.

## TECHNICAL FOUNDATION

The GWAS browser is written in the R programming language using packages that provide wrappers around popular javascript web applications including shiny (*RStudio Inc, 2013*) and rCharts (*Vaidyanathan, 2013*). Because of this, the browser can be run locally with only R and any modern web browser. Internal data processing makes use of the plyr package (*Wickham, 2011*). The javascript plots are drawn using Highcharts (highcharts.com), which is available for use under the Creative Commons Attribution-NonCommercial 3.0 License. Tables are generated using the JavaScript library Datatables (datatables.net) and xtable (*Dahl, 2013*). All of the tools and software used are either free or open source. The use of R to build the web application makes it more easily accessible to bioinformaticians to extend than if it was written in pure javascript. Many GWAS programs are written in R (*Kang et al., 2008*; *Segura et al., 2012*; *Lipka et al., 2012*). So, many scientists performing GWAS will already have some familiarity with R constructs, even if they are not computational biologists. This familiarity will hopefully make it easier for the community who is using the browser to extend it and modify it for their purposes.

## LIMITATIONS

The browser takes a fundamentally different approach from current state of the art browsers. It is focused on the ability to quickly plot a variety of GWAS experiments on a single Manhattan plot. A caveat to this ability, however, is that it cannot plot every SNP in a genotype dataset. Due to memory, time, and plotting constraints the current browser is limited to approximately 5,000 data points per trait, which is significantly less than most genotype datasets. Of course, only the most strongly associated SNPs are typically of interest, so this problem can be easily mitigated by trimming the input file to contain only significant associations (e.g., $p < 0.05$). Currently, the browser will automatically trim the number of points being plotted to only display the top 5,000 points based on the $y$-axis column. Future improvements to the browser could support the plotting of more information by binning points when zoomed out to a point where over plotting is an issue and only loading individual data points asynchronously when the zoom level is sufficient to see individual points.

The generality of the browser allows for it to be used with any SNP dataset. Only chromosome number and base pair information needs to be provided for each SNP. However, this means that specific information about the genotype dataset being used, such as minor allele frequency or linkage disequilibrium information, cannot be displayed on the plot. Of course, the flexibility of the browser would make it easy to build personalized solutions that could display additional information for specific SNP datasets, and additional tracks could be added to display linkage disequilibrium decay around the selected SNP.

One obvious extension of the browser that would address many of the limitations listed above would be to connect it to a database designed to quickly and efficiently handle all of the data that goes into a GWAS experiment. Database support would allow custom subsetting of entire GWAS datasets and if the GWAS genotype files are available, then summary data about each particular SNP could also be displayed. This would allow the browser to be incorporated into a much larger ecosystem that could take a GWAS experiment from phenotypic dataset, through running a GWAS experiment, to final analysis and visualization.

While the limitations identified above may constrain the use of the browser for certain applications, there are a number of use cases that are enabled by its current functionality. Using open source tools and GitHub for the code distribution, the browser functionalities can be enhanced by the authors or by other members of the user community.

### Funding

This work was funded by the US National Science Foundation (IOS-1126950), the US Department of Energy (DOE-SC-008796), and the US Department of Agriculture—Agricultural Research Service (5070-21000-039-00D). The funders had no role in study design, data collection and analysis, decision to publish, or preparation of the manuscript.

### Grant Disclosures

The following grant information was disclosed by the authors:

US National Science Foundation: IOS-1126950.
US Department of Energy: DOE-SC-008796.
US Department of Agriculture — Agricultural Research Service: 5070-21000-039-00D.

## Competing Interests

The authors declare there are no competing interests.

## Author Contributions

- Greg R. Ziegler conceived and designed the experiments, performed the experiments, wrote the paper, prepared figures and/or tables, performed the computation work, reviewed drafts of the paper.
- Ryan H. Hartsock performed the experiments, wrote the paper, prepared figures and/or tables, performed the computation work, reviewed drafts of the paper.
- Ivan Baxter conceived and designed the experiments, wrote the paper, reviewed drafts of the paper.

## Data Deposition

The following information was supplied regarding the deposition of related data:

GitHub: https://github.com/baxterlabZbrowse/ZBrowse

Download ZBrowse: http://www.baxterlab.org/#!/cqi0

Download ZBrowse manual:

http://media.wix.com/ugd/52737a_2a65d0deb3bd4da2b5c0190c0de343ca.pdf

For support, contact: http://baxterlabZbrowse@danforthcenter.org.

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
