# Peer review of "Zbrowse: an interactive GWAS results browser"

_PeerJ Computer Science, doi:10.7717/peerj-cs.3_

## Round 0.1 · original submission · Minor Revisions

The comments provided by the reviewers appear to me to be fair and appropriate. To these comments, I would like to add a few suggestions aimed at increasing the utility of the paper:

1. The multi-trait feature is interesting, but not well displayed in the figures. Would it be possible to provide a screen shot displaying results for more than one trait?

2. The contribution of this paper would be more clear if it were contrasted with prior work in GWAS visualization, including effort such as Locus Zoom (http://www.ncbi.nlm.nih.gov/pubmed/20634204), LocusTrack (http://www.ncbi.nlm.nih.gov/pubmed/25750659), GWASCentral (http://www.ncbi.nlm.nih.gov/pubmed/24301061), and Postgwas (http://www.ncbi.nlm.nih.gov/pubmed/23977141).

3. The paper as written contains little in the way of evaluation. Please consider adding a brief description of a case study of how the tool might facilitate interpretation of GWAS results. This might be accomplished by reframing your description of features into a narrative describing how those tools might be used to interpret a specific data set.

Reviewer 1 ·

Basic reporting

No Comment

Experimental design

Overall a great tool with an intuitive interface that facilitates analysis of SNP data over multiple traits.

More attention should be given to the type of input data / formats / columns. When evaluating the utility, I think it would be helpful to know how the data must be wrangled to use your tool. The difficulty becomes more apparent when you talk about adding additional organisms. This should be documented and/or explained as it is often the most difficult part of preparing for analysis and I think will help with additional adoption of your toolchain.

I think using R as a starting point is great and makes it instantly accessible to scientists. I think that using the web-browser as an interface makes a lot of sense. I think that longer-term it might make more sense to allow it be run out of a more global server (e.g., tomcat, jetty) to better facilitate data sharing.

In you limitations, you mention limiting the genotype data and getting around that by trimming the data. I think that a better solution (architecture allowing) would be to calculate an intermediate view for the denser traits, or automatically apply an adjustable filter.

I wholeheartedly agree with database support so long as you can keep the simplicity of your current setup for individual users.

Fig. 1 Interface: Would it be possible to show the NSP's in their own column instead of including redundant information in the name (e.g., chromosome, BP)? Would prefer just to see G/A instead (even if the name is stored the same internally).

Validity of the findings

No comments.

Additional comments

Is Z in ZBrowse for Ziegler?

Reviewer 2 ·

Basic reporting

No Commnts

Experimental design

Ok

Validity of the findings

Ok

Additional comments

In this paper, the authors have developed an R package for displaying GWAS results, along with genomic annotations of several plant genomes, including arabidopsis thaliana,maize, soybean, and sorghum. The results can also be viewed by using a browser locally.

Comparing to some existing tools, an important feature from this tool is that it can show results from multiple traits. However, the current input format requires all p-values from different traits in one column and the program cannot accept other type of formats, such as p-values of one trait in one column, which might be more common in practice.

Some other limitations:

- It can only plot about 5000 points per trait due to memory, time and plotting constraints. Given that many SNP arrays normally have hundreds of thousands SNPs, it requires users first to perform filtering to only keep small p-values.

- User interface can be further improved. For example, it is better to have an indication (i.e. waiting mouse cursor) for tasks requiring long time.

Overall, this is a simple, but potentially useful tool for some plant biologists.

---

## Round 0.2 · accepted · Accept

Thank you for addressing the concerns raised by the reviewers and in my initial letter. These changes make the paper much stronger.